# The Influence of Stand Structure on Understory Herbaceous Plants Species Diversity of *Platycladus orientalis* Plantations in Beijing, China

Ranran Cui [1], Shi Qi [1,*], Bingchen Wu [2], Dai Zhang [1], Lin Zhang [1], Piao Zhou [1], Ning Ma [1] and Xian Huang [1]

[1] School of Soil and Water Conservation, Beijing Forestry University, Beijing 100083, China
[2] Jiangxi Academy of Water Science and Engineering, Nanchang 330029, China
* Correspondence: qishi@bjfu.edu.cn; Tel.: +86-135-2204-6290

**Abstract:** Species diversity is a crucial index used to evaluate the stability and complexity of forest ecosystems. Studying the relationship between stand structure and understory herbaceous plants species diversity is useful for managers to formulate the best forest structure optimization method with the goal of improving herbaceous species diversity. In this research, *Platycladus orientalis* plantations in Beijing were taken as the research object. Pearson's correlation analysis was used to explore the single-factor correlation between stand structure and understory herbaceous plants species diversity; furthermore, a typical correlation analysis and multiple linear regression were used to explore the multi-factor correlation and analyze the dominant stand structure parameters affecting understory herbaceous plants species diversity. In the range of stand structures studied, the results showed that canopy density was negatively correlated with the Shannon–Wiener index and Simpson index ($p < 0.01$), and tree density was negatively correlated with the Shannon–Wiener index ($p < 0.05$). In terms of stand spatial structure, the mingling degree was positively correlated with the Shannon–Wiener index, Simpson index, Margalef richness index and Pielou evenness index ($p < 0.05$), while the uniform angle was negatively correlated with the Pielou evenness index ($p < 0.05$). The correlation coefficient of the first group of typical variables in the typical correlation analysis was 0.90 ($p < 0.05$); from this group of typical variables, it can be concluded that canopy density is the most influential indicator affecting the comprehensive index of understory herbaceous plants species diversity, with a load of $-0.690$, and the Shannon–Wiener index and Simpson index are the most responsive indicators of changes in the comprehensive index of stand structure, with loads of 0.871 and 0.801, respectively. In the process of the management of *Platycladus orientalis* plantations under a low altitude, south slope, thin soil layer and hard soil parent material, in order to improve the herbaceous species diversity, the canopy density of the overstory and tree density should be appropriately reduced. Additionally, it is necessary to regulate the horizontal spatial structure of stands. When the trees are randomly distributed and the mingling degree is high, the species diversity of herbs can be increased.

**Keywords:** stand structure; understory vegetation; species diversity; canonical correlation analysis





## 1. Introduction

Species diversity has long been a prevalent issue in the area of forest ecology [1–3], and species diversity, representing one of the key indicators used to evaluate the complexity and stability of forest ecosystems, and it can help to maintain the function of the forest ecosystem. Many studies have shown that the increase in species diversity can improve the stability and herbaceous plants productivity of plantations [4,5]. As an important component of forest communities, understory vegetation plays an indispensable role in improving the physical and chemical properties of soil, preventing soil erosion, and maintaining ecosystem functions [6,7]. The understory herbaceous plants species diversity

is influenced by factors such as stand structure, site conditions and climate, among which the variability of water, heat and light, and soil nutrients in understory vegetation caused by different stand structures is a key factor affecting the understory herbaceous plants species diversity on a small scale [8,9]. Therefore, it is important to investigate factors of stand structure that affect understory herbaceous plants species diversity.

Stand structure is divided into the following two parts: the stand non-spatial structure and stand spatial structure. Currently, most of the research on the factors affecting species diversity still focus on environmental factors [10–13], and relatively few studies have been conducted on the effects of stand spatial structure and non-spatial structure on species diversity. When environmental factors do not seriously disturb the community, stand structure influences the number and composition of species in the understory. The stand structure indices such as canopy density and tree density have an effect on species richness and diversity [14,15]. It is important to understand how stand structure affects understory herbaceous plants species diversity, as it is of great significance for the rational management and nurturing of plantations. Stand spatial structure is a description of the spatial pattern between trees within a forest ecosystem, reflecting spatial characteristic information of stands in the arbor layer in the horizontal and vertical directions [16], and it is more focused on reflecting the spatial distribution characteristics of stands. The stand spatial structure cannot only determine the competition intensity between neighboring trees, but also the spatial niche among trees [17]. Changes in the spatial structure characteristics of the arbor layer contribute to the spatial heterogeneity of the stand; thus, the understory vegetation habitat changes, which ultimately has an impact on the understory herbaceous plants species diversity [18,19]. Therefore, exploring the reasonable stand structure is of great significance for improving the species diversity of the herbaceous layer, enhancing the stability of plantations and allowing the functions of forests to proceed.

As an ecological construction project, the Beijing–Tianjin Sand-storm Source Control II Project is of great strategic significance in China [20]. Its main aim is to protect and improve vegetation coverage, and the project adjusts stand structure as a way to improve the understory herbaceous plants species diversity. Following the implementation of the Beijing–Tianjin Sand-storm Source Control II Project from 2013 to 2021, the total area of the project in Beijing now amounts to about 732.5 km$^2$, of which the area of *Platycladus orientalis* plantations is about 380.08 km$^2$, accounting for 52% of the total area of the project. The *Platycladus orientalis* plantations of the project are mainly located in low altitude, south slope, thin soil layer and hard soil parent material areas, because *Platycladus orientalis* tends to grow towards the sun in a low altitude, and it can adapt to the barren environment. In this study, in order to explore methods of forest management which can be used to effectively improve the understory herbaceous plants species diversity, 16 sample plots located in areas of a low altitude, south slope, thin soil layer and hard soil parent material were selected based on the Beijing–Tianjin Sand-storm Source Control II Project to investigate the relationship between the stand structure and understory herbaceous plants species diversity. The aim was to provide a basis for the adjustment of the stand structure of *Platycladus orientalis* plantations so as to improve the stability of the plantations.

## 2. Materials and Methods

### 2.1. Overview of the Study Area

The study area was located in the mountainous area of Beijing (Figure 1), and its geographical coordinates are 39°28′~41°05′ N, 115°25′~117°30′ E. The area has a warm, temperate semi-humid and semi-arid monsoon climate, with four distinct seasons. Its average annual temperature is 13–14 °C, with a frost-free period of 180–200 days, and annual average precipitation of 470–655 mm. Along the elevation gradient, from high to low, the soil types are Eutric Cambisols, Chromic Cambisols and Gleyic Cambisols. The vegetation types are mainly coniferous forests and broad-leaved forests, among which *Pinus tabuliformis* and *Platycladus orientalis* are the dominant species in coniferous forests. The forest in this study belongs to *Platycladus orientalis* plantations, the soil is Chromic

Cambisols, and the tree species mainly include *Platycladus orientalis*, *Armeniaca sibirica* and *Cotinus coggygria*.

**Figure 1.** Overview of the study area.

### 2.2. Sample Plot Setting

The Beijing–Tianjin Sand-storm Source Control II Project mainly includes low-quality and inefficient forests within its transformation project, alongside a closed forests project and difficult site afforestation project. The forests selected in this study were closed from 2018 to 2021, and were middle-aged and near-mature forests planted in the last century through tree planting. The closure measures mainly include setting up closure fences, closure signs and arranging forest patrollers, while forestry measures adopted artificial means to promote natural regeneration. Of the 80 sub-compartments in the same condition, 20% were selected to set up sample plots. The 16 sample plots were established in 2021 in the sub-compartments of the closed forest of the project. Based on the results of available forestry surveys, sample plots with a similar altitude, slope, aspect, soil thickness and soil texture were selected. Sample plots were located in areas with a low altitude (100–300 m), south slope (SS), thin soil layer (0–30 cm) and hard soil parent material. Sixteen standard plots with *Platycladus orientalis* as the dominant tree species were selected, and the accompanying tree species were *Armeniaca sibirica* and *Cotinus coggygria*, with an area of 20 m × 20 m. Each tree in the sample plot was measured, and the investigation factors included tree species composition, DBH, tree height, the coordinates of each tree and other factors. Among them, canopy density was measured using the sample line method, the sample line was set along the diagonal of the sample plot to calculate the total length of the canopy falling on the sample line, and the canopy density was equal to the ratio of

the total length of the canopy to the total length of the sample lines. The tree density can be obtained from the data of the forest survey. The study procedures are as follows: set five 1 m × 1 m herbaceous samples in the corners and the middle of each plot, and record the species names, the number of plants and the topographic information of each plot, including altitude, slope and aspect. The basic information of each sample plot is provided in Table 1.

**Table 1.** Basic information of each sample plot.

| Plot | Altitude (m) | Slope (°) | Aspect | Tree Height (m) | DBH (cm) | Canopy Density | Tree Density (Stem·ha$^{-1}$) | Age (a) | Proportion of Dominant Tree Species (%) |
|------|--------------|-----------|--------|-----------------|----------|----------------|-------------------------------|---------|------------------------------------------|
| 1 | 272 | 23 | SS | 6.1 | 7.1 | 0.8 | 1125 | 49 | 56 |
| 2 | 223 | 16 | SS | 5.2 | 7.6 | 0.5 | 1000 | 45 | 50 |
| 3 | 203 | 24 | SS | 4.0 | 6.7 | 0.6 | 975 | 45 | 97 |
| 4 | 297 | 19 | SS | 4.8 | 9.6 | 0.6 | 1175 | 60 | 72 |
| 5 | 120 | 16 | SS | 7.4 | 11.0 | 0.8 | 1500 | 80 | 57 |
| 6 | 181 | 22 | SS | 3.6 | 6.0 | 0.6 | 1575 | 40 | 65 |
| 7 | 225 | 21 | SS | 7.2 | 10.9 | 0.7 | 1425 | 78 | 79 |
| 8 | 285 | 24 | SS | 4.0 | 7.2 | 0.7 | 1100 | 53 | 61 |
| 9 | 166 | 25 | SS | 7.1 | 8.0 | 0.8 | 2125 | 69 | 93 |
| 10 | 125 | 17 | SS | 7.4 | 12.1 | 0.7 | 1300 | 80 | 85 |
| 11 | 155 | 24 | SS | 4.9 | 9.6 | 0.5 | 1550 | 65 | 89 |
| 12 | 100 | 18 | SS | 6.2 | 8.4 | 0.8 | 1175 | 50 | 68 |
| 13 | 160 | 22 | SS | 4.8 | 9.0 | 0.5 | 1300 | 52 | 81 |
| 14 | 229 | 22 | SS | 4.6 | 6.5 | 0.6 | 1250 | 34 | 56 |
| 15 | 199 | 24 | SS | 5.5 | 8.9 | 0.8 | 1275 | 76 | 96 |
| 16 | 239 | 23 | SS | 3.9 | 6.1 | 0.6 | 1200 | 36 | 96 |

*2.3. Research Methods*

2.3.1. Stand Structure Parameters

Stand structure is mainly divided into stand spatial structure and stand non-spatial structure parameters. According to the coordinates and DBH of trees in the sample plot, the method of Winklemass [21] and Excel were used to calculate the stand spatial structure parameters in various plots.

In this study, in terms of stand non-spatial structure parameters, the following four indices were adopted: tree height, DBH, canopy density and tree density, and in terms of stand spatial structure parameters, the following four indices were adopted: uniform angle, neighborhood comparison, mingling degree and opening degree. The stand spatial structure is based on the stand spatial structure unit. Each tree in the forest and its adjacent trees (n) can form a stand spatial structure unit. To avoid the adjacent trees of the object tree on the boundary falling outside the sample plots and thus affecting the results, a 5 m buffer zone was set, the tree in the buffer zone could not be set as the object tree, and the number of adjacent trees n was 4. The schematic diagram of the sample plot is shown in the Figure 2.

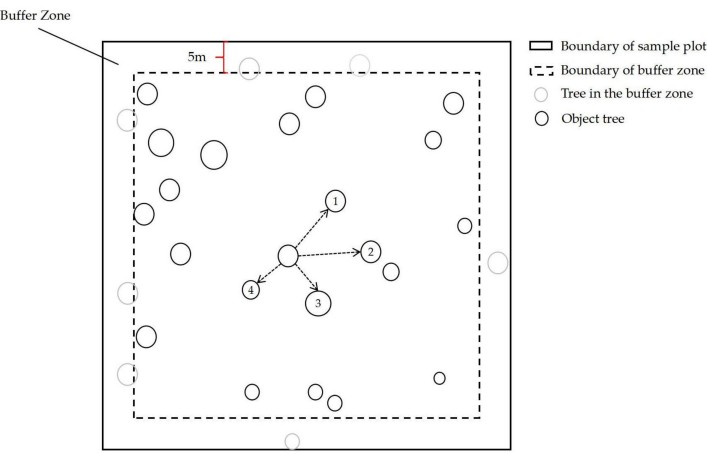

**Figure 2.** Schematic diagram of the sample plot. Note: Adjacent trees n = 4, Buffer zone = 5 m.

The formula and the meaning of each parameter are as follows:

(1) Uniform angle is an index reflecting the horizontal spatial distribution pattern of trees [22], and it was calculated according to Equation (1):

$$W_i = \frac{1}{n} \sum_{j=1}^{n} Z_{ij} \tag{1}$$

where $W_i$ is the uniform angle of object tree i, n is the number of adjacent trees. When the angle of adjacent trees is less than the standard angle, $Z_{ij} = 1$; otherwise, $Z_{ij} = 0$, and the standard angle is $360°/(n + 1)$.

(2) Neighborhood comparison is an index reflecting the degree of differentiation of the object tree (DBH, tree height and crown width, etc.) [23], in order to express the spatial dominance of the stand, and it was calculated using Equation (2):

$$U_i = \frac{1}{n} \sum_{j=1}^{n} K_{ij} \tag{2}$$

where $U_i$ is the neighborhood comparison of object tree i, and n is the number of adjacent trees. When the DBH of the object tree i is smaller than the adjacent tree j, $K_{ij} = 0$; otherwise, $K_{ij} = 1$.

(3) The mingling degree is an index reflecting the degree of isolation between trees [24], and it was calculated using Equation (3):

$$M_i = \frac{1}{n} \sum_{j=1}^{n} V_{ij} \tag{3}$$

where $M_i$ is the mingling degree of the object tree i, and n is the number of adjacent trees. When the object tree i is not the same species as the adjacent tree j, $V_{ij} = 1$; otherwise, $V_{ij} = 0$.

(4) The opening degree is an index reflecting the light transmission conditions in the forest [25], and it was calculated using Equation (4):

$$K_i = \frac{1}{n} \sum_{j=1}^{n} \frac{D_{ij}}{H_{ij}} \tag{4}$$

where $K_i$ is the opening degree of object tree i, n is the number of adjacent trees. $D_{ij}$ is the horizontal distance between the object tree i and the adjacent tree j, and $H_{ij}$ is the tree height of the adjacent tree j.

### 2.3.2. Indices of Understory Herbaceous Plants Species Diversity

In this study, the Simpson index, Shannon–Wiener index, Margalef richness index and Pielou evenness index [26] were selected to reflect the understory herbaceous plants species diversity.

(1) Margalef richness index

The Margalef richness index is an index reflecting the number of species in a community, and it was calculated using Equation (5):

$$R' = \frac{S-1}{\ln N} \tag{5}$$

(2) Simpson index

The Simpson index, indicates the probability that two individuals are randomly selected from the community and not put back, and that two herbaceous individuals belong to the same species. The greater the probability that two herbaceous individuals

belong to the same species, the higher the concentration of herbaceous individuals and the lower the species diversity. It was calculated by using Equation (6):

$$D = 1 - \sum_{i=1}^{s} P_i^2 \tag{6}$$

(3)  Shannon–Wiener index

The Shannon–Wiener index applies an information-theoretic approach to predict which species the next collected individual belongs to; and if the community species diversity is higher, the greater the uncertainty in predicting the next individual is. It was calculated by using Equation (7):

$$H' = - \sum_{i=1}^{s} P_i \ln P_i \tag{7}$$

(4)  Pielou evenness index

The Pielou evenness index reflects the distribution of individuals of all species in a community, and the higher the species evenness index, the more evenly the number of individuals of each species is distributed. It was calculated by using Equation (8):

$$J_{sw} = - \sum_{i=1}^{s} \frac{P_i \ln P_i}{\ln S} \tag{8}$$

where S is the number of species, N is the total number of individuals of all species, and $P_i$ is the ratio of the number of the ith species to the overall number of species.

2.3.3. Data Processing

In this paper, we adopted the Pearson correlation analysis, multiple linear regression and typical correlation analysis to explore the relationships between stand structure and understory herbaceous plants species diversity with SPSS.

(1)  Pearson's correlation analysis

The Pearson correlation coefficient [27] was used to calculate the univariate correlation between stand structure and understory herbaceous plants species diversity with the following Equation (9):

$$r = \frac{\sum(x - \bar{x})(y - \bar{y})}{\sum(x - \bar{x})^2 \sum(y - \bar{y})^2} \tag{9}$$

where r is correlation coefficient, x and y are samples, and $\bar{x}$ and $\bar{y}$ are the average of samples.

(2)  Canonical Correlation Analysis

Canonical Correlation Analysis (CCA) [28] is a mathematical method used to measure the correlation between two sets of multiple variables, where one set of variables is the stand structure indices and the other is understory herbaceous plants species diversity indices, assuming that the two sets of variables are as follows:

$$\begin{cases} X = (x_1, x_2, \ldots, x_n) \\ Y = (y_1, y_2, \ldots, y_n) \end{cases} \tag{10}$$

One or more representative comprehensive variables, $Z_i$ and $V_i$, were selected among the two sets of variables. This comprehensive variable can be used to replace the original variable. The equation is as follows:

$$\begin{cases} Z_i = a_1x_1 + a_2x_2 + \ldots + a_nx_n \\ V_i = b_1x_1 + b_2x_2 + \ldots + b_nx_n \end{cases} \tag{11}$$

The correlation between X and Y was explored using the relationship between linear combinations of $Z_i$ and $V_i$ to find the best vectors *a* and *b* to maximize the correlation coefficient between $Z_i$ and $V_i$.

(3)    Multiple linear regression

Stepwise regression was conducted to establish a multiple linear regression model with species diversity indices as the dependent variable and stand structure indices as the independent variables, and the equation was as follows.

$$y = \beta_0 + \beta_1 x_1 + \beta_2 x_2 + \ldots + \beta_p x_p + \mu \qquad (12)$$

where y is the dependent variable, $x_p$ is the independent variable, $\beta_p$ is a constant, and $\mu$ is residual.

## 3. Results and Analysis

### 3.1. Basic Characteristics of Stand Structure and Species Diversity

There were 56 herbaceous plants species in the selected sample plots, involving 26 families and 50 genera, among which *Asteraceae* was the most abundant, accounting for 27% of the total number of herbaceous plants species, followed by *Poaceae* and *Rosaceae*. In terms of herbaceous plants abundance, the top five species were *Artemisia carvifolia* (relative abundance is 8.8), *Carex hancockiana* (relative abundance is 8.6), *Setaria viridis* (relative abundance is 5.5), *Cleistogenes caespitosa* (relative abundance is 5.4) and *Chenopodium album* (relative abundance is 5.2), which belonged to *Asteraceae*, *Cyperaceae*, *Poaceae*, *Poaceae*, and *Chenopodiaceae*, respectively.

The basic information of stand structure parameters (stand non-spatial structure and stand spatial structure) and the understory herbaceous plants species diversity index sample plots is shown in Table 2, from which it can be seen that the coefficient of variation of most of the indices was large. Additionally, the coefficient of variation of the mingling degree was the largest at 67.9%. The variation in canopy density, uniform angle and neighborhood comparison was small among the sample plots, and the coefficients of variation which were 16.7%, 6.9% and 4.9%, respectively.

**Table 2.** Basic information of stand structure and species diversity index of the sampling plots.

| Category | Index | Mean ± Standard Deviation | Minimum | Maximum | Coefficient of Variation (%) |
|---|---|---|---|---|---|
| Stand non-spatial structure | Tree height (m) | 5.4 ± 1.3 | 3.6 | 7.4 | 24.6% |
| | DBH (cm) | 8.4 ± 1.9 | 6.0 | 12.1 | 21.9% |
| | Canopy density | 0.7 ± 0.1 | 0.5 | 0.8 | 16.7% |
| | Tree density (stem·ha$^{-1}$) | 1315 ± 280 | 975 | 2125 | 21.3% |
| Stand spatial structure | Uniform angle | 0.5053 ± 0.0347 | 0.4186 | 0.5391 | 6.9% |
| | Neighborhood comparison | 0.5044 ± 0.0247 | 0.4571 | 0.5379 | 4.9% |
| | Mingling degree | 0.3075 ± 0.2087 | 0.0278 | 0.6667 | 67.9% |
| | Opening degree | 0.5945 ± 0.2747 | 0.2683 | 1.4281 | 46.2% |
| Species diversity | Shannon–Wiener index | 0.8220 ± 0.2748 | 0.2752 | 1.2636 | 33.4% |
| | Simpson index | 0.4853 ± 0.1390 | 0.1980 | 0.6342 | 28.6% |
| | Margalef richness index | 0.7537 ± 0.2678 | 0.3913 | 1.3815 | 35.5% |
| | Pielou evenness index | 0.7825 ± 0.1602 | 0.3970 | 0.9518 | 20.5% |

### 3.2. Univariate Correlation Analysis between Stand Structure and Understory Herbaceous Plants Species Diversity

The univariate correlation analysis between stand structure and understory herbaceous plants species diversity was performed using the Pearson correlation coefficient. The result, as shown in Figure 3, indicates that tree height and DBH were not significantly correlated

with understory herbaceous plants species diversity ($p > 0.05$). In Figures 3 and 4, in the range of the stand structure studied, the canopy density was significantly negatively correlated with the Shannon–Wiener index and Simpson index ($p < 0.01$), and tree density was significantly negatively correlated with the Shannon–Wiener index ($p < 0.05$). In terms of stand spatial structure, the neighborhood comparison and opening degree were not significantly correlated with understory herbaceous plants species diversity ($p > 0.05$). The mingling degree was significantly positively correlated with all four indices, the Shannon–Wiener index, Simpson index, Margalef richness index and Pielou evenness index ($p < 0.05$), and the uniform angle was significantly negatively correlated with the Pielou index ($p < 0.05$).

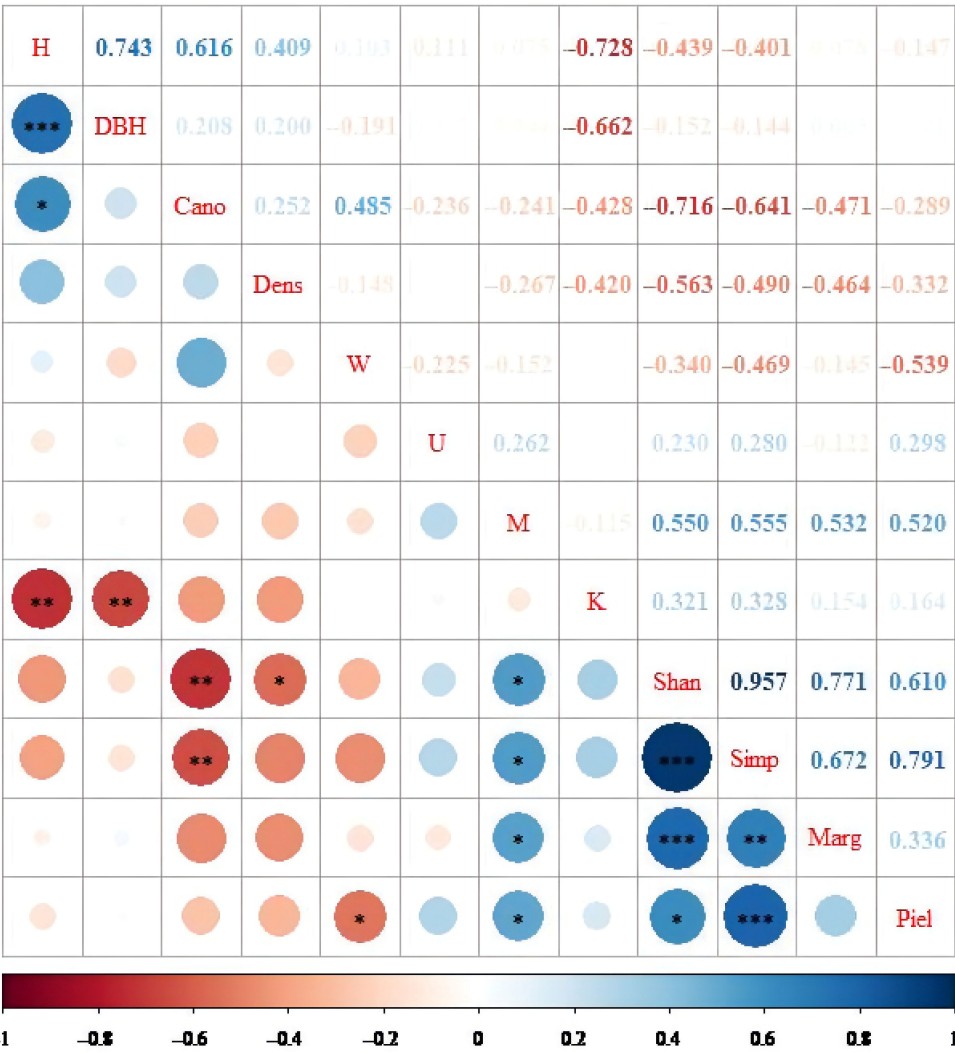

**Figure 3.** Correlation coefficients between stand structure and understory herbaceous plants species diversity. Note: ***: $p < 0.001$; **: $p < 0.01$; *: $p < 0.05$. H—Tree Height; Cano—Canopy density; Dens—Tree density; W—Uniform angle; U—Neighborhood comparison; M—Mingling degree; K—Opening degree; Shan—Shannon–Wiener index; Simp—Simpson index; Marg—Margalef richness index; Piel—Pielou evenness index.

### 3.3. Typical Correlation Analysis between Stand Structure and Understory Herbaceous Plants Species Diversity

The four stand structure indices that significantly influenced the understory herbaceous plants species diversity index were derived from a Pearson correlation analysis, namely the canopy density, tree density, uniform angle and mingling degree. The four stand structure indices were considered as one set of variables ($U_1$), and the four species

diversity indices were considered as another set of variables ($V_1$). The linear relationship between the two sets of variables was obtained by performing a typical correlation analysis.

It is shown in Table 3 that the typical correlation coefficient between the first group of typical variables was 0.907 and the results passed the typical correlation coefficient test at the $\alpha = 0.05$ level, indicating that there was a strong correlation between the two groups of variables; therefore, the group of variables consisting of the four indices of stand structure could be used to explain the group of understory herbaceous plants species' diversity variables. The correlation coefficients between the typical variables of the second, third and fourth group were 0.724, 0.453 and 0.137. The four indices (canopy density, tree density, uniform angle and mingling degree) were considered in all three groups; however, all three groups of typical variables did not pass the significance test. Therefore, only the first group of typical variables was analyzed.

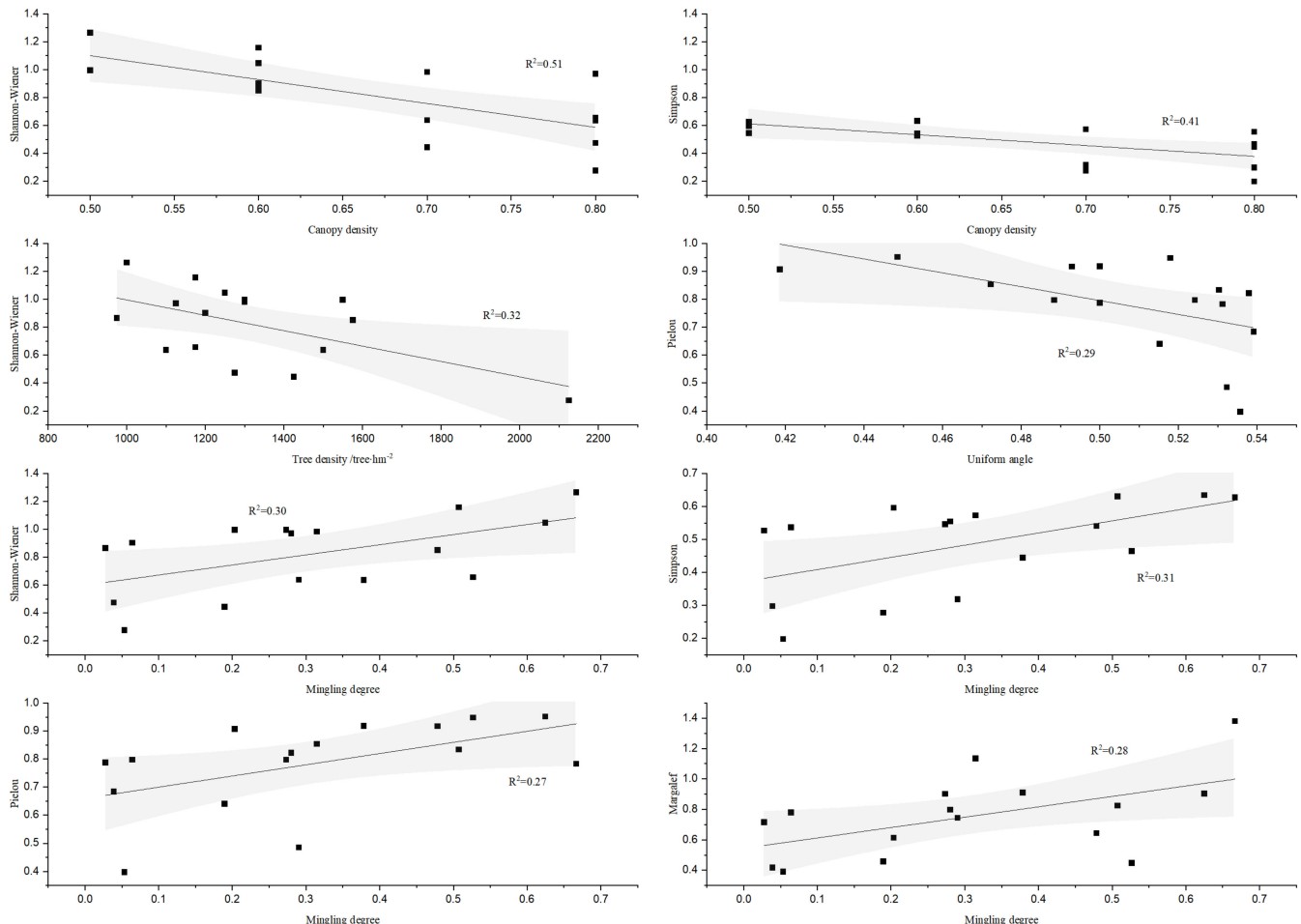

**Figure 4.** Correlation plots for cases that had a significant correlation between stand structure and understory herbaceous plants species diversity.

**Table 3.** Canonical correlation coefficient and its significance test.

| Typical Variable Group | Canonical Correlation Coefficient | *p* |
|:---:|:---:|:---:|
| 1 | 0.907 | 0.033 |
| 2 | 0.724 | 0.327 |
| 3 | 0.453 | 0.627 |
| 4 | 0.137 | 0.656 |

As can be seen from the standardized coefficients in Table 4, the equation for the first group of typical variables was as follows:

$$\begin{cases} Z_1 = -0.536X_1 - 0.473X_2 - 0.022X_3 + 0.388X_4 \\ V_1 = 2.827Y_1 - 2.331Y_2 - 0.107Y_3 + 0.733Y_4 \end{cases} \tag{13}$$

**Table 4.** Standardization coefficient and cross-load coefficient of typical variables.

| Variable | | $X_1$ | $X_2$ | $X_3$ | $X_4$ | $Y_1$ | $Y_2$ | $Y_3$ | $Y_4$ |
|---|---|---|---|---|---|---|---|---|---|
| Standardization coefficient | $Z_1$ | −0.536 | −0.473 | −0.022 | 0.388 | - | - | - | - |
| | $V_1$ | - | - | - | - | 2.827 | −2.331 | −0.107 | 0.733 |
| Cross-load coefficient | $Z_1$ | −0.690 | −0.644 | −0.246 | 0.587 | - | - | - | - |
| | $V_1$ | - | - | - | - | 0.871 | 0.801 | 0.682 | 0.525 |

Note: $X_1$—canopy density; $X_2$—Density; $X_3$—Uniform angle; $X_4$—Mingling degree; $Y_1$—Shannon–Wiener index; $Y_2$—Simpson index; $Y_3$—Margalef richness index; $Y_4$—Pielou evenness index.

In the linear combination $Z_1$, the loads of $X_1$ (canopy density) and $X_2$ (tree density) were larger, at −0.536 and −0.473, respectively, indicating that two stand non-spatial structure indices, canopy density and tree density, were the dominant factors in the comprehensive index of stand structure. In the linear combination $V_1$, $Y_1$ (Shannon–Wiener index) and $Y_2$ (Simpson index) had larger loads of 2.827 and −2.331, respectively, indicating that the Shannon–Wiener index and Simpson index played a decisive role in the comprehensive index of understory herbaceous plants species diversity. Additionally, $Y_3$ (Margalef richness index) had the smallest load of −0.107, indicating that the species richness index had a small weight in the understory herbaceous plants species diversity comprehensive index.

From the cross-loading coefficients in Table 4, it can be seen that the loads of $X_1$ (canopy density), $X_2$ (tree density) and $X_4$ (mingling degree) in $Z_1$ were larger, at −0.690, −0.644 and 0.587, respectively. The loads of $Y_1$ (Shannon–Wiener index) and $Y_2$ (Simpson index) in $V_1$ were the largest, at 0.871 and 0.801, respectively. It indicated that the canopy density and tree density of the stand non-spatial structure played a key role in affecting understory herbaceous plants species diversity, the mingling degree of stand spatial structure had a greater influence on understory herbaceous plants species diversity, and the Shannon–Wiener index and Simpson index are the most sensitive to the changes to the stand structure.

*3.4. Multiple Stepwise Regression Results*

Regression equations for each species diversity index were obtained using stepwise regression, reflecting the stand structure indices which were significantly associated with understory herbaceous plants species diversity, and four main explanatory variables affecting understory herbaceous plants species diversity were retained. Each regression equation passed the significance test ($p < 0.05$).

As can be seen from Table 5, among the stand non-spatial structure indices, the regression equation of the Shannon–Wiener index ($R^2 = 0.765$) retained two factors, canopy density and tree density, indicating that canopy density and tree density played a key role in affecting the Shannon–Wiener index. The regression equation of the Simpson index ($R^2 = 0.581$) retained the canopy density with its standardized coefficient of −0.539, which had a considerable effect on the Simpson index. Canopy density and tree density were excluded from the stepwise regression equations of the Margalef richness index and Pielou evenness index. Tree height and DBH were excluded from the stepwise regression equations of all species' diversity indices, indicating that the effect of DBH and tree height on the understory herbaceous plants species diversity was small.

Among the stand spatial structure indices, the mingling degree could be used in the regression equations of the Shannon–Wiener index, Simpson index, Margalef richness index and Pielou evenness index, indicating that the mingling degree had a significant effect on the diversity, richness and evenness of understory vegetation species. The uniform angle

was chosen in the regression equation of the Pielou evenness index ($R^2 = 0.486$), indicating that the uniform angle index influenced the Pielou evenness index, and the standardized coefficient of the uniform angle in the regression equation was $-0.471$. The neighborhood comparison and the opening degree were excluded from the regression equations of all species diversity indices, indicating that the species diversity indices were not correlated with the neighborhood comparison and the opening degree.

**Table 5.** Stepwise regression equation of species diversity index.

| Species Diversity Index | Regression Equation | $R^2$ | $p$ | AIC |
|---|---|---|---|---|
| Shannon–Wiener | y = $-0.552 \times$ Canopy $- 0.336 \times$ Density $+ 0.327 \times$ M | 0.765 | 0.000 | $-16.176$ |
| Simpson | y = $-0.539 \times$ Canopy $+ 0.425 \times$ M | 0.581 | 0.004 | $-8.951$ |
| Margalef | y = $0.532 \times$ M | 0.283 | 0.034 | $-2.353$ |
| Pielou | y = $-0.471 \times$ W $+ 0.448 \times$ M | 0.486 | 0.013 | $-5.687$ |

Note: Canopy—Canopy density; Density—Tree density; W—Uniform angle; M—Mingling degree. AIC is the akaike information criterion.

## 4. Discussion

As an important part of forest ecosystem communities, understory vegetation plays an indispensable role in plantations due to its high rate of nutrient accumulation and biological turnover; furthermore, intense competition exists between trees and understory vegetation [29,30]. The study concluded from the univariate correlation analysis, typical correlation analysis and multiple linear regression that the canopy density and tree density of the stand non-spatial structure were significantly and negatively correlated with understory herbaceous plants species diversity. The factors of canopy density and tree density had large correlation coefficients with the linear combination $V_1$ (species diversity variable group), of $-0.690$ and $-0.644$, respectively, indicating that the combined effect of canopy density and tree density on understory herbaceous plants species diversity was the largest. The coefficients of the canopy density in equations of the Shannon–Wiener index and Simpson index were $-0.552$ and $-0.539$, respectively. The main reason was that understory vegetation can receive less light when the canopy density is high; therefore, the species diversity is limited by light conditions. Some studies [31,32] showed that there is a decrease in understory herbaceous plants species diversity with increased canopy density, which is consistent with the results of this paper. The coefficient of tree density in the regression equation of the Shannon–Wiener index was $-0.336$, indicating that tree density was negatively correlated with species diversity in the herbaceous layer. Some studies also found that the number of understory vegetation species increased with rising tree density until a certain threshold [15,33]. The main reason is that when the tree density exceeds the ideal density, a competitive situation between stands forms for subsistence resources, which inhibits the growth and development of understory vegetation, resulting in a decrease in diversity. The results indicate that the range of tree density selected exceeded the threshold.

The heterogeneity of the stand spatial structure led to the formation of different community composition, light and soil conditions in the understory. The opening degree is the vertical spatial structure parameter of the stand in the study, and the opening degree was not significantly correlated with the understory herbaceous plants species diversity index in the Pearson correlation analysis ($p > 0.05$), which is consistent with the results of Zhu et al. [21]. The uniform angle, neighborhood comparison and mingling degree were all spatial structure parameters on the horizontal plane, among which the neighborhood comparison had no significant effect on understory herbaceous plants species diversity ($p > 0.05$). The uniform angle was only significantly and negatively correlated ($p < 0.05$) with the Pielou evenness index with a correlation coefficient of $-0.539$, and a correlation coefficient of $-0.471$ in the Pielou index regression equation. The uniform angle reflected the horizontal distribution pattern of the stand. The stand distribution was random when the uniform angle was small, and it was aggregated when the uniform angle was large [34]. When the distribution of trees was aggregated, the size of the forest gap

varied, leading to a decrease in species evenness. The uniform angle affects the size and location of the forest gap in the tree layer, which directly affects the value of ultraviolet radiation [35]. The appropriate forest gap size helps maintain the forest microclimate and understory microenvironment and facilitates the improvement of understory species evenness. In contrast, when the forest gap is too large, it also leads to a reduction in environmental heterogeneity within the forest gap, which results in a decrease in species evenness. The study showed that the mingling degree was significantly and positively correlated with all four understory herbaceous plants species diversity indices ($p < 0.05$), and the mingling degree was incorporated into the regression equation of each herbaceous species diversity index as an independent variable. Additionally, in the results of the typical correlation analysis, the loading coefficient of the mingling degree was 0.587 in the variable group of the stand structure index, which was the index with the greatest influence on the comprehensive index of understory herbaceous plants species diversity in terms of the stand spatial structure. The effect of the mingling degree on understory herbaceous plants species diversity was reflected in the competition among trees, leading to mutual suppression between trees and the release of nutrient space for understory vegetation. Both understory vegetation and *Platycladus orientalis* are shallow-rooted plants, while *Armeniaca sibirica* and *Cotinus coggygria* both have well-developed root systems [36,37]. Root systems of different lengths crisscross the soil, which improves the permeability and looseness of the soil, activates soil microbial activities, and thus accelerates the circulation of soil nutrients [38]. The distribution characteristics of soil nutrients explained the uneven distribution of understory vegetation [39,40]. At the same time, the increase in the mingling degree improved the heterogeneity of the forest environment, which was more conducive to the improvement of the uniformity of understory vegetation species.

The interrelationship between stand structure and understory herbaceous plants species diversity can be better reflected by the description of one-to-one, one-to-many, and many-to-many relationships in the study. In the later stage of forest management work, an improvement of the understory herbaceous plants species diversity can be achieved by adjusting the canopy density and tree density in the stand non-spatial structure, and the horizontal spatial structure (uniform angle and mingling degree) in the stand spatial structure. The three models used in this study are linear models, which cannot accurately predict the relationship between stand structure and understory herbaceous plants species diversity. To further study the response of understory herbaceous plants species diversity to stand structure, we can start from the perspective of nonlinear models and increase the number of samples in the future.

## 5. Conclusions

Stand structure had a significant effect on understory herbaceous plants species diversity, among which the canopy density and tree density had a considerable effect on the understory herbaceous plants species diversity in terms of the stand non-spatial structure, and the uniform angle and mingling degree had a great effect on understory herbaceous plants species diversity in terms of stand spatial structure. In order to increase understory herbaceous plants species diversity to enhance the plantation ecosystem function, in the subsequent management of *Platycladus orientalis* plantations, the canopy density and tree density of the forest can be improved to provide more sunlight, nutrients and space for herbaceous species. The canopy density of the overstory and tree density should be appropriately reduced. Within the range of 0.5–0.8, canopy density can be set at the value of 0.5; in the range of tree densities studied, the value of around 1000 tree·hm$^{-2}$ was found to be the most advantageous. In terms of the stand spatial structure, when trees are randomly distributed and the mingling degree is large, understory herbaceous plants species diversity is higher. Therefore, the horizontal spatial structure of the stand should be adjusted by increasing the mingling degree and disrupting the distribution pattern of trees. The results have certain limitations, however, due to the narrow range of stand structures studied and the use of a linear model. Nonlinear models can be used to investigate the

correlation between stand structure and understory herbaceous plants species diversity with increased precision in the future.

**Author Contributions:** Conceptualization, S.Q.; methodology, R.C.; validation, R.C.; formal analysis, R.C.; investigation, R.C., B.W., D.Z., L.Z., P.Z., N.M., X.H.; writing—original draft, R.C.; writing—review and editing, R.C.; visualization, R.C.; supervision S.Q.; funding acquisition S.Q. All authors have read and agreed to the published version of the manuscript.

**Funding:** This research was funded by the Forestry Monitoring and Evaluation of the 2020 Beijing–Tianjin Sand-Storm Source Control II Project, grant number 2020-SYZ-01-17JC05.

**Data Availability Statement:** Not applicable.

**Acknowledgments:** The authors are thankful to the Beijing Gardening and Greening Bureau for providing forestry engineering design data.

**Conflicts of Interest:** The authors declare no conflict of interest.

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
