# Peer review of "The Influence of Stand Structure on Understory Herbaceous Plants Species Diversity of Platycladus orientalis Plantations in Beijing, China"

_forests, doi:10.3390/f13111921_

Round 1
Reviewer 1 Report
The topic of the study is relevant, I believe that it is important to study the relationship between stand structure and species diversity of understory vegetation in different geographical locations for different forest types. However, these studies have been carried out only on 16 sample plots and a limited range of stand densities. Furthermore, only linear models were tested, which may significantly limit the implementation of the results obtained. As presented in the discussion, the study by Zhang et al. (2016) clearly showed that the relationship between canopy density and the number of plant species was not linear. Similar (Bohn & Huth, 2017; Barbier et al., 2008; Vander-Schaaf, 2008). For example, the authors found that tree density was significantly negatively correlated with herbaceous species diversity. This means that the lower the tree density, the higher the understory diversity. Thus, absurdly the best option for plant diversity would be zero trees on the plantation.
The authors did not state what specific criteria they used to select the sample plots, but only referred to the project (without citation). The conclusion states that plant diversity can be adjusted by changing stand density, but does not specify how the plantation is managed and what the management objective is. Again, there is a reference to the project, which the reader cannot access because there is no citation.
The manuscript is quite chaotic and difficult to follow. The same terms sometimes have different names.
Specific comments:
Line 9-34: In the abstract, the methods should be slightly expanded, while the results should be more synthesised.
Line 34-38: The name of “Beijing-Tianjin Sand-storm Source Control II Project” is not necessary. Conclusions should not relate only to the project, but be more universal.
Line 44-45: What does it mean “normal forest ecosystem function”?
Line 46: I do not understand what means “community productivity”.
Line 54-55: Again, what does this mean: “dominant stand structure factors”?
Line 67: “relationships among stands”, really AMONG?
Line 78: I believe that the name of the project should be given once (preferably with a reference to its description), there is no need to repeat it several times throughout the text.
Line 81: Specify the year of this implementation.
Line 84-88: There is no need to specify all project options if they have not been used in this study.
Line 90: There is no need for the abbreviation LSTH, as only this one variant was used in the study.
Line 93-95: What do you mean by “the later stage”. The objective should be more universal and not just refer to a specific project.
Line 103-104: Specify soil types according to World Reference Base for Soil Resources (WRB)
Line 105: “deciduous” and “broad-leaved” it is the same.
Line 110: Figure 1: replace Chinese symbols with international ones. Unnecessary horizontal stripes on the map. In the map legend the descriptions are probably swapped.
Line 115-117: Instead of using the LSTH acronym and name, describe in detail the criteria for selecting sample plots.
Line 118: What was the share of dominant tree species?
Line 120: In what year were the measurements taken? What age were the trees?
Line 126: I think the "index of" can be omitted from the table caption.
Line 126: What “SS” aspect means?
Line 128: Remove "spatial" in subtitle 2.3.1. This paragraph describes both types of parameters.
Line 131: Identify what software was used and specify the manufacturer.
Line 134: Describe how canopy density and tree density were determined.
Lines 163, 166, 176: For better clarity, create 4 points instead of 3 (for each index separately), use the same names in the titles as in the whole text: Simpson index, Shannon-Wiener index, Margalef richness index and Pielou evenness index
Line 183: Explain what “SPSS 25.0” means.
Line 197: The parameter “Ui” has already been used in Equation(2).
Line 215-219: This sentence is very long; it is better to divide it into two sentences.
Line 219: I suggest removing part of the sentence: “in terms of the stand non-spatial structure”.
Line 229-230: Figure 2: give a description of the indexes used. As only a linear model was used, I propose to add one more figure with correlation plots for the eight cases that had a significant correlation between stand structure and species diversity: Cano-Shan, Cano-Simp, Dens-Shan, W-Piel, M- Shan, M-Simp, M-Marg and M-Piel.
Line 244: Specify what variables the second, third and fourth group contain.
Line 266: Table 3: Specify what “Typical variable group” means.
Line 311-313: The study of Zhang et al. (2016) is NOT (!!!) consistent with the result of this paper.
Line 375-375: Conclusions should not relate only to the project, but be more universal. It should be made clear that the results obtained are of limited use, due to the narrow range of stand densities studied and the use of only a linear model.
Author Response
Dear professor,
Thank you for your comments, we have revised the article in the attachment.
Please see the attachment, thank you very much!!
Kind regards,
Ranran Cui

Reviewer 2 Report
Dear Authors, I have reviewed the paper " Effects of stand structure on species diversity of understory vegetation in Platycladus orientalis plantations of Beijing, China". The aims of the paper are germane with Forests topic. The paper is written with a moderate English level. The contribution of this paper to the scientific knowledge is moderate. In my opinion, there some important flaws and I suggest the corrections in the comments for the authors and also in the file attached. In particular Experimental design is not clear,it is necessary to better rewrite the section of materials and methods, including the description of the Project, there is confusion about the meaning of understory, confused with herbaceous species, citations in the text are not formatted as the journal requires

Author Response

(The authors gave the same response as above.)

Reviewer 3 Report
Dear Authors
Please note the following:
Title:
Your mean is “understory vegetation” or “understory herbaceous plants”? Please make the necessary corrections.
Abstract
I think the abstract is too long.
Line 29-30: I did not understand what you mean by this sentence “and there were uniform angle and mingling degree in Pielou evenness index regression equation.”
Line 84-88: In my opinion, there is no need to mention these.
Line 97: Please indicate the age of the plantation, and how it is managed, silvicultural method, and its history.
Line 117: What is your mean by “sunny slopes (SS)”?
Line 117: Please explain how was the sampling method?Why were 16 sampleplots selected?Can the results obtained from the number of 16 plots be generalized to the entire region?And how were their positions chosen? What was sampling method?
Table 1: Please write the unit of "tree density" as "stem ha -1 ".
Line 189: Please give reference for Eq. 9.
Line 190: Please give reference for the CCA.
Line 204: Please explain the Eq. 12.
Line 205: Results
Line 205: Results: In my opinion, first mention the number of species, their scientific name, their family name, and their abundance, so that the reader of the article can get a better description of the studied area.
Figure 2: Please explain the meaning of the following phrases: Cano, Dens, W, U. M, K, Shan, Simp, marg, and Piel.
Table 2: In the index column, refer to the unit of each index.Tree height (m), DBH (cm), Canopy density (%), Tree density (stem/ha)
Table 5: Please explain the “Canopy”, “Density”, “M”, “W”, and AIC“.
Line 373: There is no need to write “LSTH”
Line 372-376: Please correct it.What you mean by the words "management", "changing", "living conditions", and “living space” is not clear.
Line 378-379: Please make this sentence clearer.
Author Response

(The authors gave the same response as above.)

Round 2
Reviewer 1 Report
The authors have significantly improved the manuscript, taking into account all my comments and suggestions. I believe that in its present form it is suitable for publication in the FORESTS.
Reviewer 2 Report
Dear Authors, the effort made on the paper was appreciable. With the changes made the paper is improved